# In Situ Formation of Monohydrocalcite in Alkaline Saline Lakes of the Valley of Gobi Lakes: Prediction for Mg, Ca, and Total Dissolved Carbonate Concentrations in Enceladus' Ocean and Alkaline-Carbonate Ocean Worlds

**Keisuke Fukushi [1],\*** , **Eigo Imai [2]** , **Yasuhito Sekine [1,3]** , **Takuma Kitajima [4]** ,
**Baasansuren Gankhurel [4,5]** , **Davaadorj Davaasuren [5] and Noriko Hasebe [1]**

[1] Institute of Nature and Environmental Technology, Kanazawa University, Kakuma, Kanazawa 920-1192, Japan; sekine@elsi.jp (Y.S.); hasebe@staff.kanazawa-u.ac.jp (N.H.)

[2] School of Natural Systems, College of Science and Engineering, Kanazawa University, Kakuma, Kanazawa 920-1192, Japan; Hvf1987@outlook.jp

[3] Earth-Life Science Institute (ELSI), Tokyo Institute of Technology, 2-12-1IE-1, Ookayama, Tokyo 152-8550, Japan

[4] Division of Earth and Environmental Sciences, Graduate School of Natural Science and Technology, Kanazawa University, Kakuma, Kanazawa 920-1192, Japan; takuma.sizen@gmail.com (T.K.); gbaasnsrn@gmail.com (B.G.)

[5] Department of Geography, School of Art & Sciences, The National University of Mongolia, Ulaanbaatar 210646, Mongolia; davaadorjd@gmail.com

\* Correspondence: fukushi@staff.kanazawa-u.ac.jp; Tel.: +81-76-264-6520

**Abstract:** The nature of mineral precipitations in terrestrial alkaline soda lakes provides insights into the water chemistry of subsurface oceans on icy bodies in the outer solar system. Saturation analyses of terrestrial alkaline lakes have shown that the solution chemistries of lake waters are generally controlled by the presence of monohydrocalcite (MHC) and amorphous Mg-carbonate (AMC). However, direct observations of the formation of these metastable carbonates in natural alkaline lakes have been limited. This study provides evidence of in situ MHC formation in alkaline lakes, based on the water chemistry and mineralogy of suspended matter in Olgoy, Boon Tsagaan, and Orog Lakes (Valley of Gobi Lakes, Mongolia). The solution chemistries were close to saturation with respect to MHC and AMC, consistent with other alkaline lakes worldwide. Suspended matter was separated by the ultracentrifugation of lake water following freeze-drying. Our results show that MHC is the common mineral phase in the suspended matter. These observations confirm that MHC is the direct authigenic product of evaporation in alkaline lakes. The carbonate fraction in suspended matter from Olgoy Lake has a Mg/Ca ratio of 0.4, suggesting the formation of AMC in association with MHC. Based on the dissolution equilibria of AMC and MHC, we predict the $Mg^{2+}$, $Ca^{2+}$, and total dissolved carbonate concentrations in Enceladus' ocean to be ~1 mmol/kg, ~10 μmol/kg, and 0.06–0.2 mol/kg, respectively, in the presence of AMC and MHC. We propose that the measurements of Mg contents in plumes will be key to constraining the total dissolved carbonate concentrations and chemical affinities of subsurface oceans on Enceladus and other alkaline-carbonate ocean worlds.

**Keywords:** alkaline lake; monohydrocalcite; suspended matter; Valley of Gobi Lakes; geochemical modeling; Enceladus' ocean

## 1. Introduction

The multiple ocean worlds in the outer solar system are icy bodies with subsurface liquid oceans beneath their icy crusts [1,2]. The water chemistries of subsurface oceans on icy bodies that formed from ultramafic (e.g., chondritic) rocks beyond the $CO_2$ snowline [3] should be saline, carbonate-rich, and have high pH [4–8], suggesting $Na^\pm$, $Cl^-$, $CO_3^{2-}$, and $HCO_3^-$ as major dissolved species. Such icy bodies include Saturn's moon Enceladus in the present day, icy dwarf planets (e.g., Ceres and Pluto), and the parent bodies of D-type asteroids in the past [1,2]. The ubiquity of dissolved $Na_2CO_3$ or $NaHCO_3$ in alkaline lakes on Earth suggests that these lakes can serve as terrestrial analogues for studying the aqueous chemistry and mineral precipitations possibly occurring within ocean worlds. Understanding the processes controlling the water chemistry of terrestrial alkaline lakes will therefore allow quantitative prediction of the water chemistry of extraterrestrial liquid water.

It has been considered that the water chemistry of terrestrial alkaline lakes is determined by the evaporation and the subsequent salts formation [9,10]. Previously, the authigenic carbonate mineral species formed during evaporation in alkaline lakes were believed to be calcite or aragonite, common stable calcium carbonates [9–11]. In fact, geochemical modeling has assumed the formation of calcite and magnesite to reconstruct the water chemistries of Enceladus, Ceres, and icy planetesimals [4–8], because these carbonates are indeed found in carbonaceous chondrites, and possibly on Ceres [2,12]. However, Fukushi and Matsumiya [13] reviewed the water chemistries of a wide range of terrestrial alkaline lakes, and found that the solution chemistries of alkaline lakes are commonly and significantly oversaturated with respect to the common stable carbonate minerals. Fukushi and Matsumiya [13] conducted the solubility measurements of monohydrocalcite (MHC) and amorphous Mg carbonate (AMC) from the mixed solutions of $Na_2CO_3$, $MgCl_2$ and $CaCl_2$ at 25 °C, and obtained the constant ionic activity products (IAPs), with respect to $CaCO_3 \cdot nH_2O$ and $MgCO_3 \cdot nH_2O$ from the initial solutions, with different solution compositions. They assumed that the obtained IAPs correspond to the solubilities of MHC and AMC, respectively, and showed that the water chemistries of the alkaline lakes are close to saturation with respect to MHC and AMC. This finding suggests that the carbonate phases initially formed directly from the lake water in alkaline lakes must be MHC and AMC, which are subsequently transformed into calcite and magnesite, respectively, after sedimentation. The occurrence of this process would have affected the water chemistry on ocean worlds. However, very few reports have documented the presence of MHC and AMC in terrestrial alkaline lakes. Among the 24 alkaline lakes with pH > 9 reviewed by Fukushi and Matsumiya [13] (Table A1 in Appendix A), the formation of MHC was observed in sediments of only two lakes: East Basin Lake, Australia [14] and Walker Lake, USA [15]. To the best of the authors' knowledge, the presence of MHC has been reported in seven other saline lakes from Lake Issyk-Kul in Kyrgyzstan [16], Fellmongery Lake and Butler Lake in southern Australia [17], Manito Lake in Canada [18], Nam Co in Tibet [19] and Tsagan-Tyrm Lake [20] and Namshi-Nur Lake [21], in the west Baikal region. Moreover, no reports have shown the in situ formation of MHC in the water column of alkaline lakes. Here, we provide evidence for the in situ formation of MHC in terrestrial alkaline lakes, based on analyses of the water chemistries and the solid mineralogical compositions of the suspended matter in lake water from three alkaline lakes: Olgoy, Boon Tsagaan, and Orog Lakes (Valley of Gobi Lakes, central Mongolia).

## 2. Study Area

Olgoy, Boon Tsagaan, and Orog Lakes are located in the Valley of Gobi Lakes, central Mongolia (Figure 1). The Valley of Gobi Lakes is a latitudinally oriented depression of tectonic origin between the Khangai and Altai mountain ranges. This area receives 50–100 mm mean annual precipitation, 10–50 mm/month during the wet season (July–September), and the mean annual air temperature is 1–2 °C [22]. Olgoy Lake is a small (surface area 1.55 km$^2$, depth ~1 m [23]) closed-basin lake without inflow or outflow drainages. Boon Tsagaan and Orog Lakes are two major lakes in this area. Boon Tsagaan Lake (surface area 250 km$^2$, maximum depth 10 m) is fed by the Baidrag River, and Orog Lake (surface area 140 km$^2$, maximum depth 2 m) by the Tuin River [22]; neither has outflow drainage.

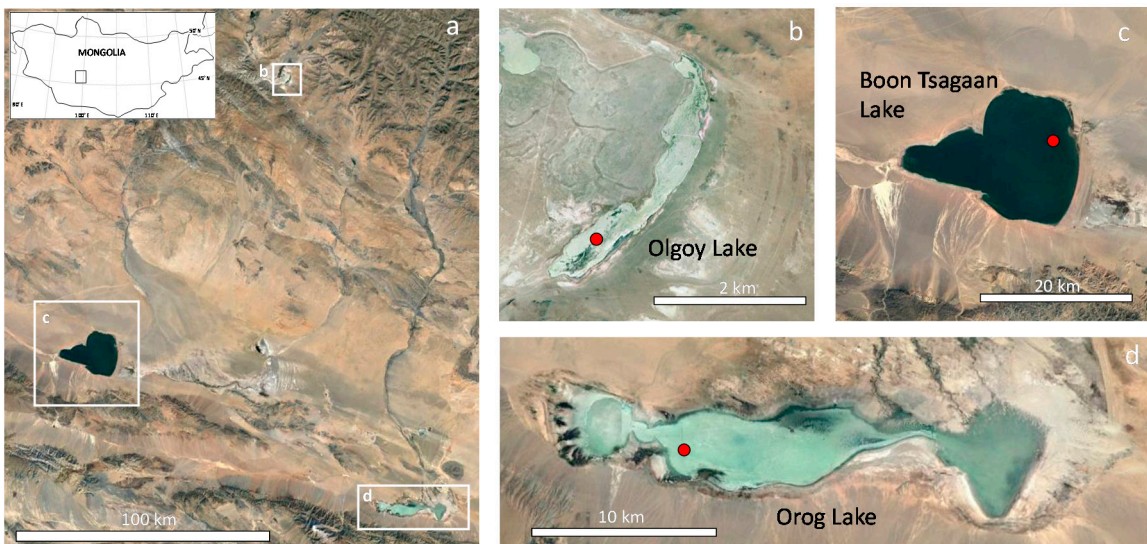

**Figure 1.** Google Earth images of (**a**) the Valley of Gobi Lakes (location in Mongolia provided in the inset map), (**b**) Olgoy Lake, (**c**) Boon Tsagaan Lake, and (**d**) Orog Lake. Red circles in (**b**,**d**) indicate the sampling points in each lake.

## 3. Materials and Methods

Water samples from Olgoy, Boon Tsagaan, and Orog Lakes were collected in August 2015 (Figure 1). The lake waters of Olgoy and Orog were turbid and were greenish and whitish in color, respectively, whereas that of Boon Tsagaan was relatively transparent. Water containing suspended matter was sampled at varying depths using a water sampler (Miyamoto Riken Ind. Co., Ltd., Osaka, Japan). Water samples were stored in 200 mL or 500 mL plastic bottles that had been pre-washed with diluted $HNO_3$ and rinsed with deionized water in the laboratory. Separate water samples for water chemistry analyses were collected at the same sampling points, filtered through 0.45-μm membranes to remove suspended matter, and the filtrate was stored in three 50-mL pre-washed plastic bottles. The first bottle was used for on-site alkalinity measurements with sulfuric acid, the second was acidified with ultrapure-grade $HNO_3$ for major cation measurements, and the third was kept for major anion measurements. The temperature, pH, electrical conductivity (EC), and oxidation–reduction potential (ORP) of the waters at the time of sampling were measured on site, using pH, EC, and ORP meters with electrodes (portable electrical conductivity pH/ORP meter, WM-32-EP; TOA-DKK Corp., Tokyo, Japan). The pH electrode was calibrated with three pH buffer solutions (pH 4.01, 6.86, and 9.18 at 25 °C) prior to measurement. Dissolved oxygen (DO) concentrations were also measured with a DO meter on site (DO-31-3IP; TOA-DKK Corp., Tokyo, Japan), and corrected for the elevation of each lake. Benthic sediments of same sampling locations were collected with a plastic ladle at Olgoy and Orog Lakes, and with the water sampler at Boon Tsagaan Lake.

Major cation concentrations (Na, K, Ca, and Mg) of the filtered water samples were analyzed by inductively coupled plasma optical emission spectroscopy (ICP-OES; ES-710S, Varian, Inc., Palo Alto, CA, USA), after adequately diluting the water samples with a 0.6% $HNO_3$ matrix. Major anion concentrations (Cl and $SO_4$) were analyzed by high-performance liquid chromatography (8020 Series, TOSOH Corp., Tokyo, Japan), after adequate dilution the deionized water. Speciation analyses of the water chemistry were performed using the REACT program of the Geochemist's Workbench and the "thermo.v8.r6+.dat" thermodynamic database [24]. The activity coefficients were calculated using the Helgeson, Kirkham, and Flowers version of the extended Debye–Hückel equation [25]. Because the lake waters were sampled at 15–25 °C, calculations were performed, assuming a temperature of 25 °C, consistent with previous speciation analyses of alkaline lakes by Fukushi and Matsumiya [13].

Suspended matter was separated from the unfiltered water samples by ultracentrifugation (Kubota 6200, Kubota Corp., Tokyo, Japan), at 15,000 rpm for 10 min. The collected suspended matters and

the benthic sediments were freeze-dried and stored in a desiccator for further analyses. A portion of each suspended-matter and sediment sample was powdered and mounted on a non-reflective sample holder for mineralogical analysis by X-ray diffraction (XRD; Ultima IV, Rigaku Corp., Tokyo, Japan; Cu Kα, 40 kV, 30 mA). Selective extractions using Morgan's solution (pH 5 acetate buffer solutions) [26,27] were performed to estimate the Mg, Ca, and Fe contents of the carbonate fraction of suspended matter from Olgoy and Orog Lakes; this was not possible for the Boon Tsagaan samples, due to the limited amounts of solid particles collected. The extractions were performed by treating 50-mg solid samples with 20 mL of 1 mol/L sodium acetate adjusted to pH 5, using acetic acid on a mix rotator for 16 h at room temperature. After extraction, the solids were separated from the suspension by ultracentrifugation at 15,000 rpm for 10 min and the supernatants were filtered through 0.45-μm membranes. The filtrates were adequately acidified and diluted with 0.6% $HNO_3$ solutions for Ca, Mg, and Fe analyses by ICP-OES.

## 4. Results and Discussion

### 4.1. Water Chemistry

The water chemistries of the Olgoy, Boon Tsagaan, and Orog Lakes are reported in Table 1. The absolute percentages of the electric charge imbalance (E.B.) of all samples (calculated by REACT) were less than 6%. The pH and salinity of each lake exceeded 9 and 1000 mg/kg, respectively, meeting the criteria of alkaline saline lakes [13]. Olgoy Lake exhibited the highest pH (~9.5) and lowest salinity (~1700 mg/kg) of the three lakes, whereas Boon Tsagaan Lake exhibited the lowest pH (~9.0) and highest salinity (~7800 mg/kg). Orog Lake was intermediate between the other two lakes, with pH ~9.1, only slightly higher than that of Boon Tsagaan, and a salinity of 3300 mg/kg. These pH and salinity ranges are moderate to low compared to other alkaline lakes worldwide (Table A1). The water chemistries of surface waters in shallow Olgoy and Orog Lakes were very similar to those of their bottom waters (0.5 m depth), although ORP and DO of bottom waters are slightly lower than those of surface water. ORP and DO concentration decreased with depth in Boon Tsagaan Lake, whereas other parameters such as pH, alkalinity, and major component concentrations were approximately the same at all depths. The depth gradient of ORP and DO is comparable for all three lakes.

Major cation concentrations in the examined lakes are always Na > Mg > Ca, consistent with other alkaline lakes worldwide (Table A1). The lower concentrations of Mg and Ca relative to Na result from the precipitation of carbonates during evaporation [9,10,13]. The dissolution reaction of calcium or magnesium ($Me^{2+}$) carbonates with $MeCO_3 \cdot nH_2O$ stoichiometry can be written as:

$$MeCO_3 \cdot nH_2O = Me^{2+} + CO_3{}^{2-} + nH_2O \qquad (1)$$

The corresponding mass action expression is:

$$K_{sp} = \frac{a_{Me^{2+}} a_{CO_3^{2-}} a_{H_2O}^n}{a_{MeCO_3 \cdot nH_2O}}, \qquad (2)$$

where $a_i$ represents the activity of the $i$th species. The activities of water and pure solid can be assumed to be unity. The ionic activity products (IAPs) with respect to calcium ($CaCO_3 \cdot nH_2O$) and magnesium carbonates ($MgCO_3 \cdot nH_2O$) were calculated using the activities of each species from the speciation analyses [10]. In the three lakes, the IAPs with respect to $CaCO_3 \cdot nH_2O$ ranged from −7.3 to −7.4, whereas those with respect to $MgCO_3 \cdot nH_2O$ ranged from −5.8 to −6.1. These values are within the ranges observed in other alkaline lakes (Table A1), although IAPs with respect to $MgCO_3 \cdot nH_2O$ in the Orog and Boon Tsagaan Lakes are slightly lower than the global average (−5.7 ± 0.3, 1SD) [13]. Figure 2 shows the relationship of $Ca^{2+}$ and $Mg^{2+}$ activities relative to the activity of $CO_3{}^{2-}$ in Olgoy, Orog, and Boon Tsagaan Lakes, compared to other alkaline lakes worldwide. The activities of these species plot within the trends of those from other alkaline lakes, which are considered to be determined by the

dissolution equilibria of MHC and AMC. This result indicates that the solution chemistries of Olgoy, Orog, and Boon Tsagaan Lakes are also controlled by the formation of MHC and AMC.

**Table 1.** Sampling information, solution chemistries; electric charge imbalances (E.B.); estimated activities (*a*) of $Ca^{2+}$, $Mg^{2+}$, and $CO_3^{2-}$; log IAPs with respect to $CaCO_3 \cdot nH_2O$ and $MgCO_3 \cdot nH_2O$; and suspended-matter Ca, Mg, and Fe contents (extracted using Morgan's solution) of samples from Olgoy, Boon Tsagaan, and Orog Lakes. "Alk." indicates total alkalinity and "n.a." indicates "not analyzed".

| Lake | | Olgoy | | Boon Tsagaan | | | Orog | |
|---|---|---|---|---|---|---|---|---|
| Sampling date | | 23 August 2015 | | 25 August 2015 | | | 26 August 2015 | |
| Location | | 46°33′7.65″ N | | 45°36′18.1″ N | | | 45°3′16.91″ N | |
| | | 100°5′37.82″ E | | 99°13′13.4″ E | | | 100°37′38.36″ E | |
| Altitude of water surface | | 2040 m | | 1300 m | | | 1210 m | |
| Sample depth | | surface | 0.5 m | surface | 4 m | 8 m | surface | 0.5 m |
| Temperature | (°C) | 16.1 | 15.0 | 20.2 | 17.9 | 17.7 | 24.7 | 24.7 |
| pH | | 9.46 | 9.44 | 9.05 | 9.05 | 9.01 | 9.08 | 9.08 |
| ORP | (mV) | 145 | 130 | 190 | 170 | 138 | 138 | 123 |
| DO | (mg/L) | 11.4 | 9.06 | 8.46 | 5.22 | 3.77 | 7.99 | 7.76 |
| EC | (mS/m) | 240 | 235 | 1040 | 1050 | 1050 | 470 | 476 |
| Alk. | (meq/L) | 14.4 | 14.3 | 19.2 | 19.5 | 19.7 | n.a. | 18.6 |
| Cl | (mg/L) | 201 | 235 | 1670 | 1640 | 1630 | n.a. | 659 |
| SO$_4$ | (mg/L) | 272 | 294 | 2730 | 2780 | 2840 | n.a. | 599 |
| Ca | (mg/L) | 7.03 | 7.17 | 21.5 | 20.2 | 20.1 | n.a. | 13.4 |
| K | (mg/L) | 3.5 | 16.6 | 57.7 | 66.8 | 59.7 | n.a. | 31.4 |
| Mg | (mg/L) | 142 | 150 | 245 | 247 | 247 | n.a. | 102 |
| Na | (mg/L) | 320 | 337 | 2070 | 2120 | 2110 | n.a. | 902 |
| E.B. | (%) | −2.5 | −1.5 | −5.5 | −4.4 | −5.2 | n.a. | −1.4 |
| Salinity | (mg/kg) | 1670 | 1760 | 7720 | 7810 | 7870 | n.a. | 3300 |
| I.S. | (mol/kg) | 0.029 | 0.031 | 0.13 | 0.13 | 0.13 | n.a. | 0.054 |
| Water type | | Na-HCO$_3$ | Na-HCO$_3$ | Na-Cl | Na-Cl | Na-Cl | n.a. | Na-Cl |
| log *a* $Ca^{2+}$ | | −4.38 | −4.37 | −4.00 | −4.03 | −4.03 | n.a. | −4.09 |
| log *a* $Mg^{2+}$ | | −2.74 | −2.72 | −2.75 | −2.75 | −2.75 | n.a. | −2.94 |
| log *a* $CO_3^{2-}$ | | −3.03 | −3.05 | −3.28 | −3.28 | −3.30 | n.a. | −3.20 |
| IAP $MgCO_3 \cdot nH_2O$ | | −5.77 | −5.77 | −6.03 | −6.03 | −6.06 | n.a. | −6.14 |
| IAP $CaCO_3 \cdot nH_2O$ | | −7.41 | −7.41 | −7.28 | −7.31 | −7.33 | n.a. | −7.29 |
| Morgan Ca | (mg/g) | 35 | 45 | n.a. | n.a. | n.a. | 82 | 87 |
| Morgan Mg | (mg/g) | 8.4 | 8.8 | n.a. | n.a. | n.a. | 10 | 10 |
| Morgan Fe | (mg/g) | 0.39 | 0.33 | n.a. | n.a. | n.a. | 1.6 | 1.5 |
| Mg/Ca | (mol/mol) | 0.39 | 0.32 | n.a. | n.a. | n.a. | 0.20 | 0.19 |

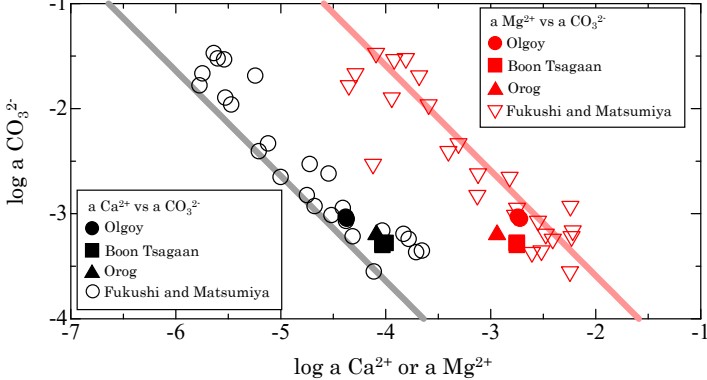

**Figure 2.** Relationship between the activity (log scale) of $CO_3^{2-}$, and that of $Ca^{2+}$ (black symbols) or $Mg^{2+}$ (red symbols) in waters from Olgoy, Boon Tsagaan, and Orog Lakes and other alkaline lakes worldwide [13]. Straight black and red lines are the solubilities of MHC and AMC, as experimentally estimated by Fukushi and Matsumiya [13].

### 4.2. Mineralogy of Suspended Matter

The XRD pattern of suspended matter in Olgoy Lake exhibited a broad hump at 20°, most likely due to organic matter (Figure 3a). Most peaks were assigned to MHC, with a few associated with quartz, plagioclase, and clay minerals, such as illite and chlorite; notably, calcite and aragonite were not detected. In contrast, the benthic sediment from Olgoy Lake was a mixture of quartz, plagioclase, illite, chlorite, MHC, and calcite (Figure 3b). Because Boon Tsagaan is the deepest of the examined lakes, we collected water samples at the surface and at 4 and 8 m depth (near the lake bottom). The amounts of suspended matter obtained at each depth were very small. Suspended solids from 4 m depth and shallower were mainly halite, which likely formed while drying the samples (Figure 3c). In contrast, suspended matter from 8 m depth contained MHC, quartz, and halite (Figure 3d); again, neither calcite nor aragonite was detected in the suspended matter. The mineralogy of the benthic sediment from Boon Tsagaan Lake was almost the same as that from Olgoy Lake (Figure 3e). Suspended matter from Orog Lake contained MHC and significant amounts of quartz, plagioclase, illite, chlorite, and calcite (Figure 3f). The benthic sediment of Orog Lake contained quartz, plagioclase, illite, chlorite, and calcite, but MHC was notably absent (Figure 3g). Except for the presence of MHC, the suspended matter exhibited the same mineralogy as the benthic sediment from Orog Lake (Figure 3h). Because Orog Lake is shallow with a large river influx, the benthic sediments were most likely resuspended in the water column.

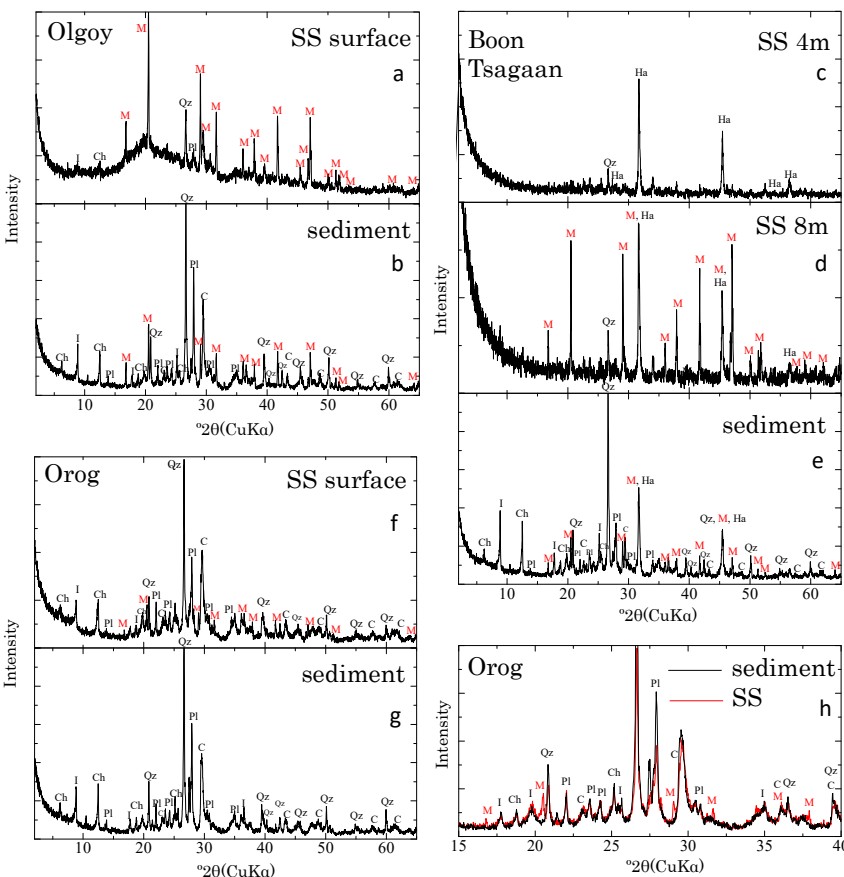

**Figure 3.** XRD patterns of suspended matter (SS) and benthic sediments from (**a**,**b**) Olgoy, (**c**,**e**) Boon Tsagaan, and (**f**,**g**) Orog Lakes. (**h**) Comparison of XRD patterns of suspended matter (red) and sediment samples (black) from Orog Lake. Mineral abbreviations: M, monohydrocalcite; C, calcite; Qz, quartz; Pl, plagioclase; I, illite; Ch, chlorite; and Ha, halite.

Previously, the authigenic calcium carbonate formed during evaporation in alkaline lakes was believed to be calcite or aragonite [10,11,28]. However, based on saturation analyses of the water

chemistries of alkaline lakes (Table A1), Fukushi and Matsumiya [13] predicted that MHC, rather than these stable calcium carbonates, controls water chemistry. This finding implies that the directly authigenic form of calcium carbonate in alkaline lakes must be MHC. Because neither calcite nor aragonite were observed in suspended matter from the examined lakes (except in the likely resuspended sediments in Orog Lake), this study confirms that MHC is the predominant calcium carbonate phase in suspended matter in alkaline lakes, evidencing the in situ formation of MHC in alkaline lakes.

The MHC was present in the suspended matter, but absent in the sediments in Orog Lake. The absence of MHC in the sediments must be caused by the transformation of MHC to calcite during deposition on the lake floor, because MHC is a metastable phase with respect to calcite [29–31]. On the other hand, the MHC was observed in the sediments as well as the suspend matters from Olgoy and Boon Tsagaan Lakes. Fukushi and Matsumiya [13] observed that a timescale of the transformation is 400–600 h under 25 °C. Munemoto and Fukushi [29] observed that the transformation rate of MHC at 10 °C is 30 times lower than that at 25 °C. The altitude of Orog Lake is significantly lower than the other lakes. The water temperature of Orog Lake is always higher than those of Olgoy and Boon Tsagaan Lakes (Table 1). The relatively high temperature in Orog Lake may explain the faster transformation of MHC. Alternatively, the different transformation rates of MHC among the three lakes are related to the presences of the trace additives in the lake waters. Yagi and Fukushi [32] showed that the presence of trace phosphate (~30 μmol/L) in the solution effectively inhibits the transformation of MHC. The lake water of Olgoy was greenish in color and rich in organic matter in the suspended matter (Figure 3a). The presence of organic matters in Olgoy Lake water suggests that the lake possesses a higher amount of the nutrients, including phosphate than other lakes. The peak intensities of MHC in the XRD patterns of the sediments from Boon Tsagaan Lake is lower than those from Olgoy Lake (Figure 3b,e). This suggests that the relative abundance of the MHC in the sediment from Boon Tsagaan Lake is smaller than that from Olgoy Lake. The possible effect of the nutrient also may explain the differences of transformation kinetics of MHC among the three lakes. The different lifetimes of MHC have the widespread implication of being not only earth environment but also extraterrestrial water body. An additional study must be conducted to elucidate the process.

The presence of suspended MHC in the three lakes in different locations and with different water chemistries suggests that the formation of MHC is likely ubiquitous in alkaline lakes; however, MHC has rarely been observed in sediments from alkaline lakes (Table A1). This lack can partly be explained by the transformation of MHC to stable carbonates. Furthermore, MHC more readily transforms to calcite with increased temperature [29]. If, in previous studies, sediments had been oven-dried before mineralogical analysis instead of the freeze-drying, MHC likely transformed to calcite while drying.

Fukushi and Matsumiya [13] predicted that the solution chemistry of alkaline lakes is also controlled by the formation of AMC. The solution chemistries of Olgoy, Boon Tsagaan, and Orog Lakes are close to saturation with respect to AMC (Figure 2). Suspended matter from Olgoy Lake contained MHC, but not calcite, aragonite, or crystalline Mg-carbonate (Figure 3a). The average Ca and Mg contents in the carbonate fraction of the suspended matter were 40 mg/g and 8.6 mg/g, respectively, a molar Mg/Ca ratio of 0.4 (Table 1), indicating the association of Mg with MHC. Nishiyama et al. [33] and Fukushi et al. [34] found that MHC formed only when the reacted solution was saturated with respect to AMC, and concluded that the formation of MHC requires the coexistence of AMC. Fukushi et al. [34] estimated the solid solubility limit of Mg in the MHC of Mg/Ca = 0.06. Recently, Kitajima et al. [35] showed that the structural incorporated Mg in MHC can only be detected from the MHC with very low Mg content (Mg/Ca < 0.01), from the X-ray absorption near edge structure analyses. Therefore, the host of Mg in suspension is likely AMC, which cannot be detected by XRD [34–36]. On the other hand, the speciation of Mg associated with MHC has still been under debate. Rodriguez-Blanco et al. [37] showed that 25% of Ca in MHC could be substituted by Mg. A recent theoretical study also confirmed a possible Mg-Ca substitution of MHC [38]. The present study cannot provide the indisputable evidence to prove the presence of AMC, because of the limited amounts of the suspended matter and the limited analytical method. AMC is also metastable phase and transforms to crystalline hydrous Mg

carbonates such as hydromagnesite and nesquehonite [34]. Small amounts of Mg-carbonates can also be overlooked from XRD patterns. The possible presence of the crystalline hydrous Mg carbonates cannot be ruled out, because the solubility of nesquehonite is similar to AMC [33]. Spectroscopic analyses are thus required to confirm the presence of AMC.

*4.3. Implications for the Water Chemistry of Enceladus' Ocean*

The evaporative concentrations and consequent carbonate precipitations within Orog, Olgoy, and Boon Tsagaan Lakes could be analogous to the concentrations of dissolved components upon the freezing of subsurface oceans of Ceres and Pluto (e.g., [2,6]). Similar concentrations of dissolved components may occur within subsurface oceans of icy moons, such as Enceladus, due to decreases in the oceanic volume upon the orbital evolution (e.g., [39,40]). The chemical components participating in the MHC and AMC formations are Ca, Mg and DIC, which are ubiquitous components and less sensitive to redox reactions, and thus applicable to the alkaline-carbonate ocean worlds. NASA's Cassini spacecraft has provided key information on the water chemistry of Enceladus' subsurface ocean via in situ analyses of Enceladus' plumes and Saturn's E ring particles derived from the plumes [1,12,41]. The results from Cassini's onboard cosmic dust analyzer (CDA) indicate that Enceladus' ocean is a Na-Cl-$CO_3$-$HCO_3$ solution with an alkaline pH of 9–11 [1,42] and mild salinity (0.05–0.2 mol Cl/kg and 0.05–0.3 mol Na/kg) [1], conditions analogous to those of terrestrial alkaline lakes (Table A1). The major cation in Enceladus' ocean is $Na^+$ (~0.3 mol/kg), followed by $K^+$ (~1 mmol/kg or Na/K ≈ 100–200) [1], whereas the concentrations of other cations, including $Mg^{2+}$ and $Ca^{2+}$, are below the CDA's detection limit of ~1 mmol/kg [1]. Results from Cassini's ion neutral mass spectrometer (INMS) indicate that the gas phase of the plume contained ~1% gaseous $NH_3$ (relative to $H_2O$ vapor) [41]; however, because of the lack of $NH_4^+$-bearing clusters in the CDA mass spectra of Saturn's E-ring particles, the $NH_4^+$ concentration may have been lower than that of $Na^+$ [1]. This suggests that the pH must be higher than the $NH_4^+$/$NH_3$ couple's $pK_a$ of ~10, assuming no enrichment or depletion of $NH_3$ relative to $H_2O$ in vapor.

CDA measurements show that ice particles in Saturn's E ring contain ~0.03 mol/kg total dissolved carbonate [1], whereas INMS measurements of the gas phase in Enceladus' plume show it to contain 0.3–0.8% degassed $CO_2$ (relative to $H_2O$ vapor) [41], corresponding to 0.2–0.4 mol/kg in solution. These high and low $CO_2$ concentrations in the gas and solid phases, respectively, may result from the degassing of dissolved $CO_2$ from liquid droplets in the source regions of the plumes, i.e., at the surface of the subsurface ocean [1]. Therefore, the actual total dissolved carbonate concentration in the ocean is likely ~0.03–0.4 mol/kg.

Based on these observations, we here discuss whether AMC and MHC might form within Enceladus' ocean, thus controlling the total dissolved carbonate concentration. If AMC forms in Enceladus' ocean, the relationship between the activities of $Mg^{2+}$ and $CO_3^{2-}$ should follow the solubility of AMC (red line in Figure 2). We calculated $a_{Mg^{2+}}$ and $a_{CO_3^{2-}}$ for Cl contents of 0.05–0.2 mol/kg at pH 9–11 [1,42] using the REACT program in the Geochemist's Workbench [24]. The calculations were conducted under 25 °C and 1 atm, because the solubility of AMC at lower temperature is currently unavailable. Na contents were calculated to satisfy the charge balance in the solutions, meaning that we treated $Na^+$ as the predominant cation in Enceladus' ocean, compared with others, such as $NH_4^+$. Therefore, this assumption provides an upper limit on the $Na^+$ concentration in the ocean. Nevertheless, the abundance of $Na^+$-bearing clusters and rarity of $NH_4^+$-bearing clusters in Saturn's E-ring particles [1] support the validity of our assumption, which would not significantly change our quantitative estimate of Na content within the accuracy of our estimates. Using the upper limit of the possible $Mg^{2+}$ content of Enceladus' plume (~1 mmol/kg) [1], the lower limits of $a_{CO_3^{2-}}$ and total dissolved carbonate concentration are obtained as $10^{-2.1}$ and 0.035 mol/kg, respectively, based on the dissolution equilibrium of AMC (Table 2). Total dissolved carbonate concentrations below 0.4 mol/kg require high oceanic pH (~10–11), due to the abundance of $HCO_3^-$ in solutions at pH ~9. To achieve $Na^+$/$Mg^{2+}$ > ~200, high Cl concentrations (0.1–0.2 mol/kg) are favorable, because the Na

concentrations are high due to the charge balance. However, Cl concentrations above 1.5 mol/kg result in Na concentrations exceeding 0.3 mol/kg, which is inconsistent with the observations. Glein et al. recently estimated the $a_{CO_2(aq)}$ from the geochemical interpretations of INMS and CDA data as $10^{-4.6}$ to $10^{-3.2}$ [8]. The $a_{CO_2(aq)}$ calculated from the pH and DIC in present study are $10^{-7.3}$ to $10^{-4.2}$, consistent with the estimation from Glein et al. [8]

**Table 2.** Thermodynamic calculations of solution chemistries equilibrated with MHC and AMC at conditions of Enceladus' subsurface ocean, as estimated from Cassini's CDA results: pH 9–11, 0.05–0.2 mol Cl/kg, and 1 mmol $Mg^{2+}$/kg (upper limit).

| | Total Cl: 0.05 mol/kg | | | | | Total Cl: 0.1 mol/kg | | | | |
|---|---|---|---|---|---|---|---|---|---|---|
| pH | 9 | 9.5 | 10 | 10.5 | 11 | 9 | 9.5 | 10 | 10.5 | 11 |
| $Mg^{2+}$ (mmol/kg) | 1.0 | 1.0 | 1.0 | 1.0 | 1.0 | 1.0 | 1.0 | 1.0 | 1.0 | 1.0 |
| $Ca^{2+}$ (mmol/kg) | 0.009 | 0.009 | 0.009 | 0.009 | 0.009 | 0.009 | 0.009 | 0.009 | 0.009 | 0.009 |
| log $a$ $Mg^{2+}$ | −3.9 | −3.7 | −3.6 | −3.5 | −3.5 | −3.9 | −3.7 | −3.6 | −3.6 | −3.6 |
| log $a$ $Ca^{2+}$ | −5.9 | −5.7 | −5.6 | −5.6 | −5.5 | −5.9 | −5.8 | −5.7 | −5.7 | −5.7 |
| log $a$ $CO_3^{2-}$ | −1.7 | −1.9 | −2.0 | −2.1 | −2.1 | −1.7 | −1.8 | −1.9 | −2.0 | −2.0 |
| DIC (mol/kg) | 1.2 | 0.2 | 0.07 | 0.04 | 0.04 | 1.3 | 0.3 | 0.10 | 0.07 | 0.06 |
| total Mg (mmol/kg) | 4 | 4 | 4 | 4 | 3 | 4 | 4 | 4 | 4 | 4 |
| total Ca (mmol/kg) | 0.06 | 0.06 | 0.06 | 0.06 | 0.06 | 0.06 | 0.06 | 0.06 | 0.06 | 0.06 |
| total Na (mol/kg) | 1.4 | 0.3 | 0.16 | 0.12 | 0.11 | 1.6 | 0.5 | 0.3 | 0.2 | 0.2 |
| $Na^+$/$Mg^{2+}$ | 1400 | 300 | 160 | 120 | 110 | 1600 | 500 | 300 | 200 | 200 |

| | Total Cl: 0.15 mol/kg | | | | | Total Cl: 0.2 mol/kg | | | | |
|---|---|---|---|---|---|---|---|---|---|---|
| pH | 9 | 9.5 | 10 | 10.5 | 11 | 9 | 9.5 | 10 | 10.5 | 11 |
| $Mg^{2+}$ (mol/kg) | 1.0 | 1.0 | 1.0 | 1.0 | 1.0 | 1.0 | 1.0 | 1.0 | 1.0 | 1.0 |
| $Ca^{2+}$ (mmol/kg) | 0.009 | 0.009 | 0.009 | 0.009 | 0.009 | 0.009 | 0.009 | 0.009 | 0.009 | 0.009 |
| log $a$ $Mg^{2+}$ | −3.9 | −3.8 | −3.7 | −3.7 | −3.7 | −3.9 | −3.8 | −3.7 | −3.7 | −3.7 |
| log $a$ $Ca^{2+}$ | −6.0 | −5.8 | −5.8 | −5.7 | −5.7 | −6.0 | −5.9 | −5.8 | −5.8 | −5.8 |
| log $a$ $CO_3^{2-}$ | −1.7 | −1.8 | −1.9 | −1.9 | −1.9 | −1.7 | −1.8 | −1.8 | −1.9 | −1.9 |
| DIC (mol/kg) | 1.3 | 0.3 | 0.13 | 0.09 | 0.07 | 1.4 | 0.4 | 0.16 | 0.11 | 0.09 |
| total Mg (mol/kg) | 4 | 4 | 4 | 4 | 4 | 4 | 4 | 4 | 4 | 4 |
| total Ca (mol/kg) | 0.06 | 0.06 | 0.06 | 0.06 | 0.06 | 0.06 | 0.06 | 0.06 | 0.06 | 0.06 |
| total Na (mol/kg) | 1.7 | 0.6 | 0.4 | 0.3 | 0.3 | 1.9 | 0.7 | 0.5 | 0.4 | 0.4 |
| $Na^+$/$Mg^{2+}$ | 1700 | 600 | 400 | 300 | 300 | 1900 | 700 | 500 | 400 | 400 |

Values in the grey background indicate inconsistencies with the observations.

Based on these calculations, the water chemistries reported in Table 3 can satisfy all observational constraints in the presence of AMC in Enceladus' ocean. The estimated total dissolved carbonate concentration is ~0.06–0.2 mol/kg, and several times that in plume ice particles (~0.03 mol/kg) [1], supporting the interpretation that dissolved $CO_2$ degasses from liquid droplets upon eruption. Based on this range of total dissolved carbonate concentrations, we predict the $Mg^{2+}$ and $Ca^{2+}$ concentrations of the ocean to be 0.6–1 mmol/kg and 5–9 µmol/kg, respectively, in the presence of AMC and MHC.

**Table 3.** Estimated pH and Na, Cl, total dissolved carbonate (as DIC), $Mg^{2+}$, and $Ca^{2+}$ concentrations in Enceladus' ocean.

| Parameter | Unit | Range |
|---|---|---|
| pH | - | 9.5–11 |
| Na | mol/kg | 0.2–0.3 |
| Cl | mol/kg | 0.05–0.18 |
| DIC | mol/kg | 0.06–0.2 |
| $Mg^{2+}$ | mmol/kg | 0.6–1.0 |
| $Ca^{2+}$ | mmol/kg | 0.005–0.009 |

Glein et al. [4] previously predicted the Mg and Ca concentrations in Enceladus' ocean, assuming equilibria with respect to calcite and Mg-bearing phyllosilicates (serpentine and talc). They estimated the Mg and Ca concentrations to be $10^{-4}$–$10^{-5}$ mmol/kg and $10^{-4}$ mmol/kg, respectively, which are about $10^{-4}$–$10^{-5}$ and $10^{-2}$–$10^{-1}$ times our estimates. This comparison demonstrates how assumptions about the reacting mineral phases in geochemical modeling can result in quite different predictions. Geochemical modeling usually considers mineral assemblages of stable rather than metastable phases. Our results demonstrate that understanding the chemical processes occurring in terrestrial environments is important to conduct the geochemical modeling for the chemical reactions in extraterrestrial conditions. To definitively conclude whether AMC and MHC or Mg-phyllosilicates and calcite control the water chemistry on Enceladus, laboratory experiments on AMC and MHC formation in the presence of Mg-phyllosilicates are required.

Our prediction that $Mg^{2+}$ concentrations are below 1 mmol/kg in Enceladus' ocean may be testable during future missions to Enceladus, via in situ plume analyses performed by an onboard high-resolution mass spectrometer [43], whereas a Mg-phyllosilicate-controlled system would produce very low $Mg^{2+}$ concentrations (~$10^{-5}$ mmol/kg) [3]. Because $CO_2$ is degassed from liquid droplets in the plume source, direct constraints on the total dissolved carbonate concentration in Enceladus' ocean would be challenging, even with in situ analyses of plume materials. Nevertheless, if the formation of MHC and AMC can be confirmed by future missions, our results suggest that the total dissolved carbonate concentration in the ocean can be determined based on the $Mg^{2+}$ concentrations of plume materials (Figure 2), which is important in estimating the precise available chemical affinities for methanogenesis in alkaline-carbonate ocean worlds such as Enceladus [41]. Furthermore, MHC and AMC are metastable phases with respect to calcite or aragonite. The lifetime of MHC at a low temperature (<10 °C) is several years based on the laboratory kinetics experiments of the MHC transformation [13,29]. The presence of MHC in the ocean worlds can constrain the occurrence of on-going variations in the seawater composition near the plume source, within the timescale of several years. On Enceladus, for instance, such variations in the seawater composition could be caused by changes in hydrothermal activities by periodic, tidal energy dissipation [44], by long-term changes in the oceanic volume owing to the orbital evolution [39,40], and by the evaporative concentration of seawater near the plume source [45]. If found, the occurrence of MHC and AMC would be suggestive of these dynamic changes in the seawater composition.

## 5. Conclusions

We investigated the water chemistries and mineralogical compositions of suspended matter in three alkaline lakes of the Valley of Gobi Lakes. The solution chemistries of the lake waters were close to saturation with respect to MHC and AMC, consistent with other alkaline lakes worldwide. XRD analyses of suspended matter in lake water from Olgoy and Boon Tsagaan Lakes showed that MHC, rather than stable calcite, is the predominant suspended carbonate. MHC was also confirmed in suspended matter from Orog Lake, but was absent in the benthic sediments, indicating the transformation of MHC to calcite during deposition on the lake floor, due to the metastability of MHC. The Mg/Ca ratio in the carbonate fraction of suspended material from the Olgoy Lake was around 0.4, suggesting the presence of a discrete Mg-bearing phase. The absence of any such crystalline phase in XRD patterns indicates the presence of AMC associated with MHC, although the indisputable evidence to prove the presence of AMC was not provided from the present study. Therefore, this study provides evidence that the water chemistries of alkaline lakes are controlled by the formation of MHC, and possibly AMC, during evaporation.

Based on our field observations, we discussed the possibility of AMC and MHC formation in the subsurface ocean of Enceladus. The presence of AMC and MHC in Enceladus' ocean is consistent with currently available observations of cryovolcanic plume compositions. We predict the $Mg^{2+}$, $Ca^{2+}$, and total dissolved carbonate concentrations in Enceladus' ocean to be ~1 mmol/kg, ~10 µmol/kg, and 0.06–0.2 mol/kg, respectively, in the presence of AMC and MHC. We suggest that measurements of the

Mg contents of plume particles will be a key observable for estimating the total dissolved carbonate concentrations and chemical affinities of subsurface oceans on Enceladus and other ocean worlds.

**Author Contributions:** Conceptualization, K.F., Y.S. and N.H.; Fieldwork, K.F., E.I., B.G., D.D. and N.H.; Experiments, E.I. and B.G.; Modeling, K.F. and T.K.; Writing—Original Draft Preparation, K.F.; Writing—Review and Editing, Y.S. and N.H.; Funding Acquisition, K.F., N.H. and Y.S. All authors have read and agreed to the published version of the manuscript.

**Funding:** Financial support was provided by Grants-in-Aid for Scientific Research from the Japan Society for Promotion of Science, grant numbers 16H05643, 17H06456, 17H06458, and 20H00195; a cooperative research program of the Institute of Nature and Environmental Technology, Kanazawa University, grant numbers 19035, 20010, and 20061; and the Functional Materials based on Mongolian Natural Minerals for Environmental Engineering, Cementitious, and Flotation Processes, grant number J11A15 subproject under the Mongolian−Japanese Engineering Education Development Project.

**Acknowledgments:** The authors thank K. Kashiwaya of Kanazawa Univ., N. Katsuta of Gifu Univ., and Y. Tanaka of Kyung Hee Univ., car drivers, and local people for help in our field surveys.

**Conflicts of Interest:** The authors declare no conflict of interest.

## Appendix A

**Table A1.** Solution chemistries, electric charge imbalances (E.B.), and log ionic activity products (IAPs) with respect to $CaCO_3 \cdot nH_2O$ and $MgCO_3 \cdot nH_2O$ of alkaline lakes worldwide [13]. DIC, dissolved inorganic carbon; I.S. denotes ionic strength. "n.a." indicates "not analyzed". Mineral abbreviations: Arg, aragonite; (Mg-)Cal, (Mg-)calcite; Dol, dolomite; Gay, gaylussite; Ika, ikaite; (H-)Mgs, (hydro)magnesite; MHC, monohydrocalcite.

| Lake | Area | pH | I.S. mol/kg | Na mmol/kg | K mmol/kg | Mg mmol/kg | Ca mmol/kg | Cl mmol/kg | SO$_4$ mmol/kg | DIC mmol/kg | E.B. | log IAP CaCO$_3 \cdot n$H$_2$O | log IAP MgCO$_3 \cdot n$H$_2$O | Carbonate Mineralogy |
|---|---|---|---|---|---|---|---|---|---|---|---|---|---|---|
| Duikou | China | 9.4 | 0.14 | 100 | | 13 | 0.23 | 97 | 7.5 | 16 | −0.8% | −7.45 | −5.62 | n.a. |
| ErquanjingII | China | 9.3 | 0.14 | 120 | | 8.3 | 0.18 | 93 | 7.7 | 20 | 0.8% | −7.53 | −5.78 | n.a. |
| Kulun | China | 9.6 | 0.17 | 160 | | 3.1 | 0.077 | 93 | 3.9 | 49 | −0.4% | −7.62 | −5.81 | n.a. |
| Qinghai | China | 9.23 | 0.26 | 170 | 3.9 | 35 | 0.30 | 170 | 24 | 19 | 0.5% | −7.53 | −5.45 | n.a. |
| Maoertu | China | 10.18 | 4.7 | 6900 | 410 | 2.8 | 0.12 | 5500 | 59 | 430 | 9.8% | −7.47 | −5.98 | n.a. |
| Huhejilin or Huhejaran | China | 10.46 | 1.9 | 1900 | 91 | 4.7 | 0.21 | 830 | 130 | 460 | 0.5% | −7.07 | −5.45 | Arg, Cal |
| Nuoertu | China | 10 | 1.6 | 1600 | 120 | 6.4 | 0.31 | 820 | 120 | 380 | −0.4% | −6.93 | −5.37 | Arg |
| Sumujilin-S | China | 10.54 | 2.2 | 2200 | 110 | 6.3 | 0.19 | 1200 | 160 | 500 | −2.0% | −7.12 | −5.33 | n.a. |
| Sumujilin-N | China | 10.64 | 2.9 | 3400 | 110 | 3.6 | 0.19 | 1200 | 290 | 660 | 8.7% | −7.11 | −5.57 | n.a. |
| Yabulai 11 | China | 9.7 | 0.29 | 250 | 7.2 | 12 | 0.099 | 190 | 27 | 31 | −1.1% | −7.65 | −5.47 | n.a. |
| Oigon | Mongolia | 9.02 | 0.40 | 320 | | 31 | 1.2 | 240 | 59 | 27 | −0.3% | −7.02 | −5.65 | no carbonates from XRD |
| Telmen | Mongolia | 9.06 | 0.11 | 67 | | 16 | 0.74 | 43 | 16 | 21 | 0.2% | −7.02 | −5.67 | no carbonates from XRD |
| Tsegeen | Mongolia | 9.07 | 0.20 | 150 | | 18 | 1.2 | 110 | 28 | 16 | −0.4% | −7.01 | −5.87 | no carbonates from XRD |
| Shaazgai | Mongolia | 9.28 | 0.23 | 220 | 2.3 | 0.59 | 0.21 | 130 | 7.8 | 67 | 0.3% | −7.25 | −6.66 | Cal |
| Khilganta | Russia | 9.8 | 0.77 | 640 | | 64 | 0.23 | 480 | 130 | 25 | −1.0% | −7.61 | −5.17 | Cal |
| Van | Turkey | 9.7 | 0.43 | 340 | 14 | 4.4 | 0.10 | 160 | 25 | 150 | −9.6% | −7.43 | −5.55 | Arg, Mg-Cal, Cal |
| Specchio di Venere | Italy | 9 | 0.31 | 280 | 10 | 5.2 | 0.15 | 230 | 7 | 63 | −3.2% | −7.57 | −5.95 | Arg, H-Mgs, Cal, Dol |
| East Basin | Australia | 9.15 | 0.85 | 720 | 15 | 50 | 0.81 | 840 | 4.6 | 33 | −3.4% | −7.19 | −5.38 | MHC, Dol, Mgs, H-Mgs, Arg, Cal, Mg-Cal |
| Richmond | Australia | 9.1 | 0.045 | 24 | 0.46 | 7 | 0.64 | 24 | 1.8 | 11 | −2.8% | −7.08 | −5.98 | n.a. |
| Walyungup | Australia | 9.1 | 0.15 | 92 | 1.8 | 23 | 0.32 | 100 | 4.9 | 9.8 | 5.9% | −7.66 | −5.80 | Arg, H-Mgs, Dol |
| Mono | U.S.A. | 9.8 | 1.54 | 1500 | 45 | 1.7 | 0.10 | 640 | 140 | 460 | −2.6% | −7.41 | −5.96 | Cal, Ika, Gay |
| Wilkinson | U.S.A. | 10.04 | 1.3 | 1000 | 193 | 1.3 | 0.08 | 470 | 220 | 250 | −5.7% | −7.55 | −6.13 | n.a. |
| Big Soda | U.S.A. | 9.7 | 0.41 | 350 | 7.9 | 6.0 | 0.12 | 180 | 58 | 66 | −4.1% | −7.45 | −5.64 | n.a. |
| Pyramid | U.S.A. | 9.39 | 0.08 | 58 | | 7.2 | 0.21 | 59 | 1.7 | 19 | −9.4% | −7.16 | −5.74 | n.a. |
| Walker | U.S.A. | 9.45 | 0.17 | 130 | 4.2 | 5.6 | 0.27 | 63 | 21 | 38 | −1.8% | −7.35 | −5.70 | MHC, Cal, Arg |

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
