# Peer review of "In Situ Formation of Monohydrocalcite in Alkaline Saline Lakes of the Valley of Gobi Lakes: Prediction for Mg, Ca, and Total Dissolved Carbonate Concentrations in Enceladus’ Ocean and Alkaline-Carbonate Ocean Worlds"

_minerals, doi:10.3390/min10080669_

Round 1
Reviewer 1 Report
Review of “In-situ formation of monohydrocalcite in alkaline saline lakes ….” Minerals July-2020
This manuscript reports findings of field sample analyses concerning suspensions and water chemistry of alkaline saline lakes in central Mongolia, and the potential implications of the results to aqueous chemistry of extraterrestrial water bodies. Other than on-site routine measurements of water properties (conductivity, pH/Eh, and so on), aqueous composition analyses were performed via ICP-OES on collected water samples. For the solid portions, xrd was the sole analytical technique used. I find the measurements were carefully performed and results well-presented and convincing. I don’t have much expertise in aqueous geochemistry of extraterrestrial water bodies, but I felt the discussions on the implications may be a bit overly stretched, perhaps out of neccessity so due to the very nature of inference to planetary science. Even if we take out the implication section, overall, I think the results remains interesting because the relevance to earthly water bodies, saline lakes or even oceanic environments. I recommend it for publication with minor revisions highlighted below:
- The detection of MHC in suspensions of three lakes is rather interesting, particularly in the context of MHC absence in associated sediments. This may have wide spread of implication to earth environment. On the one hand, this is indicative of the ephemeral nature of MHC due to its lower thermodynamic stability as well as its sensitivity to environmental changes (e.g. T and redox condition); on the other hand, it may also be a reflection of the effect from organic matters. Assuming the MHC components in the suspension undergo a phase transition to calcite, it is fascinating to see the rapid transformation given the complete absence of MHC in sediments. I wonder if the authors can provide microphotographic images of the suspended particles with MHC grains to give the readers a realistic view of this “mysterious” mineral phase. Also, it would help if some discussion is provided in terms of factors controlling the stability/transformation of MHC to the more stable crystalline minerals.
- I wonder what it means by the solubility of AMC. As far as the standard terminology goes, AMC may not be one phase because of the water content that can change depending on how the AMC was synthesized and how long it is dried. Unless there are data showing AMC is a unique and discrete phase, I think the readers would like to know what the red line in Fig 2 is indicating. I understand the data was from previous work by the same authors, but feel some sort of discussion is needed in terms of relating to condition of present study.
- One of the issues (as a non-specialist of extraterrestrial water science) I have trouble to resolve is the effect of organic matters on MHC and AMC formation when data from the current study was used to implicate outer solar system icy/water bodies. On our earth, we know AMC is often associated with biomineralization. I’m sure the lakes studies in this work have organic matters and microbes in them. That being the case, it would be important to expand the discussion to include how the bio/organic effect may play out, or provide organic composition analyzes of the water/suspension/sediments so the readers to gain a full picture when inferring extraterrestrial environmental conditions.
- The relation between MHC / AMC and the presence of Mg2+ is interesting and could use a bit discussion. For example, is the presence of Mg2+ to certain level a necessary requirement for MHC formation? Is it reasonable to speculate that AMC is not a discrete phase and Mg is actually associated with ACC or even MHC? In this regard I think some sort of SEM-EDX or EMP-WDX analyses of the suspension/sediment particles may provide clues.

Reviewer 2 Report
This paper reports the results of field investigations of the water, suspended matter, and sediment compositions of three alkaline saline lakes in the Valley of Gobi, Mongolia, to show that they are controlled by solution equilibria involving monohydrocalcite (MHC) and (perhaps) amorphous Mg-carbonate (AMC). These results are then applied to the ocean of Saturn's moon Enceladus, for which the comparatively few data available suggest a similar composition to alkaline saline lakes, to provide revised compositional estimates assuming similar control by MHC and AMC.
I find this paper well-written. Except for minor comments below, the field and modeling methods seem robust to me and the results adequately interpreted. The extrapolation to Enceladus is careful, creative, and insightful. I do have two comments (questions, really) regarding that extrapolation that could be addressed (answered) by added discussion at the end of Section 4.3. My other comments are minor. All are provided below.
1. The lack of MHC in lake Orog sediment is interpreted as conversion to stable calcite. This brings about the question of the kinetic timescale of conversion of MHC and AMC to more stable carbonates in the context of both lake Orog and Enceladus' ocean. From Munemoto & Fukushi (2008) or other pertinent work, can a timescale be estimated and used to explain the absence of MHC in sediments from lake Orog, but not the other two lakes? Extrapolating to Enceladus, could the presence or absence of MHC constrain the ocean temperature and/or timescale of core-surface transport or sedimentation?
2. Relatedly, the manuscript would benefit from a discussion of differences between Enceladus and the Gobi lakes. To what extent are compositions arising from evaporation (Gobi) analogous to compositions arising from hydrothermal circulation (Enceladus) and, perhaps, freezing (Ceres and Pluto)? Additionally, presumably a major difference between the Gobi lakes and subsurface oceans may be ORP / DO concentrations, as these lakes are in contact with Earth's oxic atmosphere. To what extent might this bear on the extrapolation to Enceladus?
Minor comments provided in order of line number:
l.48-49 "present day, and, icy dwarf planets (e.g., Ceres and Pluto)" -> "present day, icy dwarf planets (e.g., Ceres and Pluto)"
l. 106 "Surface sediments": do the authors mean "benthic sediments"? (i.e. at the bottom, not the top of the water column). Please clarify.
l. 113 "after adequately dilution the deionized water" -> "after adequate dilution of the deionized water"
l. 115 "The activity coefficient was calculated" -> "activity coefficients were calculated"
l. 121 How long were samples centrifuged and how was it determined that this was enough time? Did centrifugation take place at room temperature? Please specify. Same at l. 131.
l. 144-146: Presumably, the reason why no appreciable decrease in ORP and DO concentration is seen in Olgoy and Orog lakes is because of their very low depth. From Table 1 it appears that the depth gradient of ORP and DO concentration is about the same for all three lakes. I think this is worth mentioning.
Equation (1): Since the pH is in the stability regime of bicarbonate rather than carbonate, shouldn't the reaction be equilibrated with HCO3- rather than CO3^2-? In this case, pH would factor in to equation (2). This shouldn't change the calculated IAPs but would be formally more accurate.
l. 201-203: If "MHC is the predominant calcium carbonate phase in suspended matter in alkaline lakes", why wasn't MHC found at 4 m depth in Boon Tsagaan lake (outlier measurement)?
l. 216: "in previous studies, sediments had been oven-dried": I read this as leaving the samples dry too long. But it seems that the drying temperature also likely would play a role in letting the conversion to CaCO3 happen. It may be worth repeating here that the samples were freeze-dried.
l. 236-238: The NH4+/NH3 couple could then set an independent rough constraint on the pH of Enceladus' ocean. Since the concentration of NH4+ is not higher than ~1% by mass, which is about the concentration of NH3 assuming no enrichment or depletion of NH3 relative to H2O in the vapor, the pH must be higher than the couple's pKa of ≈10 at 0ºC (e.g. Bell et al. 2007; doi 10.1071/EN07032). Unless I am missing something, this may be worth mentioning. (Glein et al. 2015 did, albeit in passing.)
l. 250 What temperature (especially) and pressure were used in the calculations? Please specify.
l. 263 and l. 266 Typos: Change "Grein" to "Glein"
l. 283 For Ca, the estimate of Glein et al. (2015) is 10^-2 - 10^-1 times the present estimate of 0.005–0.009 mmol/kg. (Not just 10^-1.)
Reviewer 3 Report
‘In-situ formation of monohydrocalcite in alkaline saline lakes of the Valley of Gobi Lakes: Prediction for Mg, Ca, and total dissolved carbonate concentrations in Enceladus’ ocean and alkaline-carbonate ocean worlds’ by Keisuke Fukushi, Eigo Imai, Yasuhito Sekine, Takuma Kitajima, Baasansuren Gankhrel, Davaadorj Davaasuren and Noriko Hasebe.
The article is devoted to findings of calcium carbonate monohydrate (monohydrocalcite) in three alkaline lakes in Mongolia (Olgoy, Boon Tsagaan, and Orog Lakes). Based on the dissolution equilibria of monohydrocalcite (MHC) and amorphous Mg-carbonate (AMC) the authors predict the Mg2+, Ca2+, and total dissolved carbonate concentrations in Enceladus’ (Saturn’s moon) ocean. Authors used inductively coupled plasma optical emission spectroscopy and high-performance liquid chromatography for water chemistry determination and powder X-ray diffraction (PXRD) for mineralogical analysis. Despite a high level of the work, the draft have several serious problems, which should be addressed before acceptance.
- Presence of amorphous Mg-carbonate was not proved.
One could confirm the presence of AMC on PXRD pattern, however, according to author they did not found any signs of AMC. The only possible sign (a broad hump at 20°) ‘most likely due to organic matter’ (line 179). I strongly recommend to provide spectroscopic investigations to confirm the presence of AMC (as authors suggest to their selves on line 225-226).
- Possible Mg incorporation in MHC should be discussed.
According to authors (lines 309-11) ‘The Mg/Ca ratio in the carbonate fraction of suspended material from Olgoy Lake was around 0.4, suggesting the presence of a discrete Mg-bearing phase. The absence of any such crystalline phase in XRD patterns indicates the presence of AMC associated with MHC.’ However, according to Rodriguez-Blanco et al. (2014) (https://doi.org/10.1016/j.gca.2013.11.034) up 25% of Ca could be substituted by Mg. Recent studies of Chaka (2019) (https://doi.org/10.1021/acs.jpca.9b00180) also confirmed a possible Mg-Ca substitution. It seems that authors disregard all the works on that topic as it make their calculation quite complex.
- I suggest to expand review on MHC findings and discuss other saline lakes (including Issyk-Kul, where MHC was found (Semenov, E.I. (1964) Hydrous carbonates of calcium and sodium. Kristallografiya (Sov. Phys. Crystal.), 9, 109–110)). Authors also disregard works on other saline lakes from that region (e.g. Solotchina et al., 2009 (https://doi.org/10.1016/j.quaint.2009.02.027)) and other saline lakes from Transbaikalia (e.g. Solotchina et al., 2011; https://doi.org/10.1134/S1028334X11020267), where MHC was found.
Round 2
Reviewer 3 Report
Dear authors,
thank you for your answers.
I think you should clearly state that AMC was predicted, but not unequivocally proven as there is no indisputable evidence to prove the presence of AMC in your samples (-the limited analytical methods, -the metastable nature of AMC, -the limited amounts of the suspended matters for sequential extraction analyses). This should be clearly stated in all sections of the articles (results and conclusions).
I also suggest to discuss other opportunities of Mg specialization in carbonate sediments, such as small amounts of Mg-carbonates, which can be overlooked on PXRD. Moreover, one could try to estimate unit cell parameters of MHC and compare them with other MHC (e.g. Swainson, 2008; Rodriguez Blanco et al. 2014) in order to evaluate Mg content in MHC.
Best regards,
The Reviewer
